# Fluorescent artificial receptor-based membrane assay (FARMA) for spatiotemporally resolved monitoring of biomembrane permeability

Frank Biedermann [1,2✉], Garima Ghale[2], Andreas Hennig[2] & Werner M. Nau [2✉]

The spatiotemporally resolved monitoring of membrane translocation, e.g., of drugs or toxins, has been a long-standing goal. Herein, we introduce the fluorescent artificial receptor-based membrane assay (FARMA), a facile, label-free method. With FARMA, the permeation of more than hundred organic compounds (drugs, toxins, pesticides, neurotransmitters, peptides, etc.) through vesicular phospholipid bilayer membranes has been monitored in real time ($\mu$s-h time scale) and with high sensitivity (nM-$\mu$M concentration), affording permeability coefficients across an exceptionally large range from $10^{-9}$–$10^{-3}$ cm s$^{-1}$. From a fundamental point of view, FARMA constitutes a powerful tool to assess structure-permeability relationships and to test biophysical models for membrane passage. From an applied perspective, FARMA can be extended to high-throughput screening by adaption of the microplate reader format, to spatial monitoring of membrane permeation by microscopy imaging, and to the compartmentalized monitoring of enzymatic activity.

[1] Institute of Nanotechnology, Karlsruhe Institute of Technology (KIT), Hermann-von-Helmholtz Platz 1, 76344 Eggenstein-Leopoldshafen, Germany.
[2] Department of Life Sciences and Chemistry, Jacobs University Bremen, Campus Ring 1, 28759 Bremen, Germany. ✉email: frank.biedermann@kit.edu; w.nau@jacobs-university.de

The permeability of molecules through biological membranes is a fundamental physicochemical property, e.g., it allows cells to regulate the influx/efflux of nutrients, neurotransmitters, and pharmaceutical drugs, as well as of xenobiotics[1–3]. Several assays for screening membrane permeability of potentially bioactive compounds have been developed[4–9], two of which have become routine in pharmaceutical-industrial and academic settings: the parallel artificial membrane (PAMPA)[10–12] and the Caco-2 cell permeability assay[13]. The cost-efficient PAMPA assay quantifies the passive diffusion of substances through macroscopic and flat synthetic membranes, whose composition has been optimized to model the permeability behavior of phospholipid-based biomembranes[12]. The Caco-2 assay aims to identify substances that can pass through a monolayer of colon epithelial cells, which is highly relevant for the gastrointestinal uptake. Obstacles are its slow turnaround time, high cost, and the potential involvement of metabolic pathways, active transporters or efflux systems[4]. Importantly, both the PAMPA and Caco-2 set-ups are usually restricted to single-point measurements and the results depend strongly on extrinsic factors, such as the rate of stirring[11], which complicates access to elementary kinetic information.

The permeability of aromatic molecules is of pivotal importance, because aromatic moieties are ubiquitously occurring in peptides, hormones, neurotransmitters, toxins, biocides, as well as colorants and drugs. For instance, ~80% of oral drugs contain at least one aromatic moiety[14]. Although aromatic compounds are chromophoric, and in principle detectable by spectrophotometry, about 40% of all substances fail ultraviolet visible (UV/Vis) detection because of low solubility (typically 50–100 µM are required)[11] or insufficient absorbance in the near UV/Vis region[15]. Consequently, the development of a sensitive fluorescence-based permeability assay has been a long-standing goal[16]. Frequently, fluorescently labeled analytes have been used but the attachment of fluorescent tags can drastically influence the permeability characteristics[16,17]. Thus, label-free fluorescence-based permeation assays are sought for but currently limited to purpose-selected analytes. For instance, dynamic fluorescence quenching of an encapsulated reporter dye was used for permeation monitoring of redox-active analytes at very high (millimolar) concentrations[18]. We showed that an indicator displacement assay[19] can be adopted to monitor membrane translocation of a label-free highly charged antimicrobial peptide and selected charged amino acid derivatives through membrane pores[20,21]. For this purpose, an environment-responsive dye is precomplexed with a host to yield a chromophoric or emissive reporter pair. In the presence of a host-binding analyte, the dye is competitively displaced, giving rise to a quantifiable signal change. The limitations of indicator displacement assays, most notably its limited scope to strongly binding and slowly translocating analytes, are described in detail in the Supplementary Information.

Supramolecular chemists have in recent years designed or discovered several artificial hosts for biorelevant organic analytes, e.g., calixarenes, cyclodextrins, cucurbit[$n$]urils and their acyclic congeners, pillar[$n$]arenes, deep cavitands, and molecular tweezers[22–27]. Of particular interest for sensing applications are fluorescent artificial receptors (FARs) with a wide analyte scope, a high binding affinity in water, and a rapid analyte-binding kinetics[26]. We hypothesized that through encapsulation of such FARs in liposomes, a FAR membrane assay (FARMA) can be established that will be capable of monitoring the membrane passage of a very wide range of label-free, biologically relevant analytes in real time. The schematic FARMA concept is depicted in Fig. 1a. In this study, we demonstrate its utility for self-assembled FARs composed of the macrocycle cucurbit[8]uril

and a tightly bound dicationic reporter dye as the co-factor (Fig. 1b)[26]. This system is excellently suited for monitoring the permeation of aromatic analytes (>100 compounds tested, see Fig. 2) through a phospholipid bilayer membrane in the biologically most relevant nM–µM range.

## Results

**Analyte binding to fluorescent artificial receptors.** FARs were self-assembled in aqueous buffer from the macrocycle cucurbit[8] uril (CB8) and fluorescent, dicationic dyes **D1–D3**, forming discrete 1:1 CB8•dye complexes (Fig. 1a)[26,28]. These FARs possess residual space in their cavity that serves as a binding pocket for aromatic moieties, e.g., phenyl, indoyl, and naphthyl species (see Fig. 1c). Crucial for their use in FARMA, the herein utilized CB8-dye-based FARs are phospholipid membrane-impermeable and photostable (see Supplementary Information, e.g., Fig. 9).

The signal response (generally emission quenching)[26] of **FAR-1** and **FAR-2** towards selected analytes with an aromatic recognition motif was quantified in homogeneous aqueous solution by fluorescence titration experiments (see the Supplementary Fig. 1 and Supplementary Table 1). Some analytes show the appearance of an exciplex emission band when binding to **FAR-3** (see Supplementary Fig. 4). Importantly, the response is "immediate" even in stopped-flow experiments, suggesting that FARs are suitable for time-resolved ($t \geq 100$ µs) monitoring of analyte permeation through phospholipid membranes.

Our analyte test library (Fig. 2) contained more than 100 bio- and drug-relevant compounds with aromatic moieties such as phenols, anilines, indoles, naphthalenes, polyaromatic hydrocarbons, benzimidazoles, alkylated benzenes, halogenated aryl-species, quinolines, pyridines, and furans (Fig. 2). The analytes carried a wide spectrum of functional groups, ranging from electron-donating to electron-withdrawing groups, e.g., $-NR_2$, $-NH_2$, $-OR$, $-OH$, $-F$, $-Cl$, $-Br$, $-I$, $-SH$, $-OPO_3^{2-}$, $-COR$, $-CONHR$, $-COOR$, $-COOH$, $-SO_2NH_2$, $-SO_3H$, $-CN$, and $-NO_2$. Representative analytes with immediate biological relevance are aromatic amino acids (e.g., tryptophan), metabolites (tryptophanamide, TrpNH$_2$), neurotransmitters (e.g., serotonin), antibiotics (e.g., penicillin G), drugs (e.g., omeprazole), herbicides (e.g., propanil), fungicides (e.g., thiabendazole), carcinogenics (e.g., anthracene), food additives (e.g., raspberry ketone), and bioactive peptides (e.g., somatostatin). We are unaware of any alternative method, which allows the direct real-time assaying of membrane permeability of such a structurally diverse library. In fact, an adaption of the method to a dye-displacement format allows for the monitoring of additional aliphatic analytes, e.g., adamantane derivatives such as the drug memantine and alkyl amines such as the metabolite cadaverine (see Figs. 14–17 in the Supplementary Information).

**Implementation of FARs into membrane translocation assays.** Most established drug permeation assays are based on single-point determinations where the signal is recorded before and after a fixed time period. To demonstrate that FARMA can effectively complement these state-of-the-art single-point assays, the following sequence of steps was executed: (1) FARs were encapsulated into phospholipid liposomes formed in aqueous buffer by a rehydration and freeze–thaw procedure, followed by chromatographic separation of FAR-containing liposomes from non-encapsulated FARs, see the "Methods." (2) The purified FAR-containing liposome suspension was transferred to fluorescence cuvettes or into microwells and the emission spectra at $t = 0$ were recorded. (3) Aliquots of the test analyte were added and the emission spectra were recorded again after fixed time intervals (5–60 min).

**a** Fluorescent Artificial Receptor Membrane Assay **(FARMA)**

**Fig. 1 Operational principle of the fluorescent artificial receptor membrane assay (FARMA). a** Encapsulation of membrane-impermeable fluorescent artificial receptors (FARs) into liposomes spatially separates the FARs from the subsequently added analyte. Upon analyte permeation through the membrane into the liposome, and subsequent rapid analyte complexation by FAR, a readily observable change in the fluorescence intensity of the FAR can be observed (typically emission quenching). **b** Expected emission-readout for a permeable vs. an impermeable analyte. **c** Structures of the synthetic host CB8 and dyes (**D1**–**D3**), from which **FAR-1**, **FAR-2**, and **FAR-3** were assembled.

Comparison of the fluorescence spectra before and after analyte addition after a fixed time yielded a pattern that is in full accordance with the graphical depiction of the FARMA principle in Fig. 1a: a quenching of the emission intensity of the FARs was observed for permeating analytes (e.g., naphthalene or indole) (Fig. 3a, b and Supplementary Fig. 2–4), whereas non-permeating analytes (e.g., zwitterionic tryptophan) caused no significant changes in the fluorescence signal (Fig. 3c and Supplementary Figs. 2 and 3). The FARMA method is transferable to microplate reader format in disposable and cost-economic plastic wells (Supplementary Fig. 2).

Several control experiments were carried out to ensure that the observed fluorescence changes were not due to a disruption of liposomes caused by the analytes and not due to the leakage of the FARs (see Fig. 3a, b and the Supplementary Information).

**Assay sensitivity**. The series of FARMA experiments showed that an analyte concentration of 10 μM is generally sufficient to differentiate between permeating and non-permeating analytes. In fact, for selected analytes such as indole, phenol, tryptophan methyl ester (TrpOMe) and tryptamine even nM to low μM concentrations were sufficient (Figs. 3b and 4, and Supplementary Fig. 5). Converted to 96-well microplate reader format, the FARMA sensitivity corresponds to 4–400 ng/well, depending on analyte. Thus, the sensitivity of FARMA is comparable to established mass spectrometry-coupled permeation assay formats and superior to common absorbance-based permeation assay formats

requiring typically 50–100 μM analyte concentration[4,10–13]. We observed that with liposome-encapsulated FARs, analytes can be detected at least an order of magnitude more sensitively compared to experiments with non-encapsulated FARs in homogeneous solution. This sensitivity enhancement through FAR encapsulation will be also relevant for additional sensing applications besides permeation monitoring, e.g., for environmental monitoring.

**Time-resolved analyte translocation monitoring by FARMA**. When carrying out the measurements in a time-resolved manner, i.e., with a continuous recording of the emission intensity, the permeation process of the analytes through the phospholipid bilayer membrane becomes observable in real-time (see the schematics in Fig. 1b and the plotted experimental data in Fig. 3c–g). This constitutes a major advancement compared with PAMPA or Caco-2 assays. Depending on the absolute permeation rates, FARMA kinetics can be either monitored upon manual mixing with standard fluorometers or microplate readers (min–h) or by stopped-flow techniques with fluorescence detection (ms–s). The CB8-dye-based FARs are useful for monitoring of both slow and fast analyte translocation kinetics, because FAR•analyte complex formation is very rapid (up to diffusion limited[26]) and, thus, not rate-determining.

As an example of slow translocation kinetics, the permeation of tryptamine (Fig. 3d) and TrpNH$_2$ (Supplementary Fig. 6) was

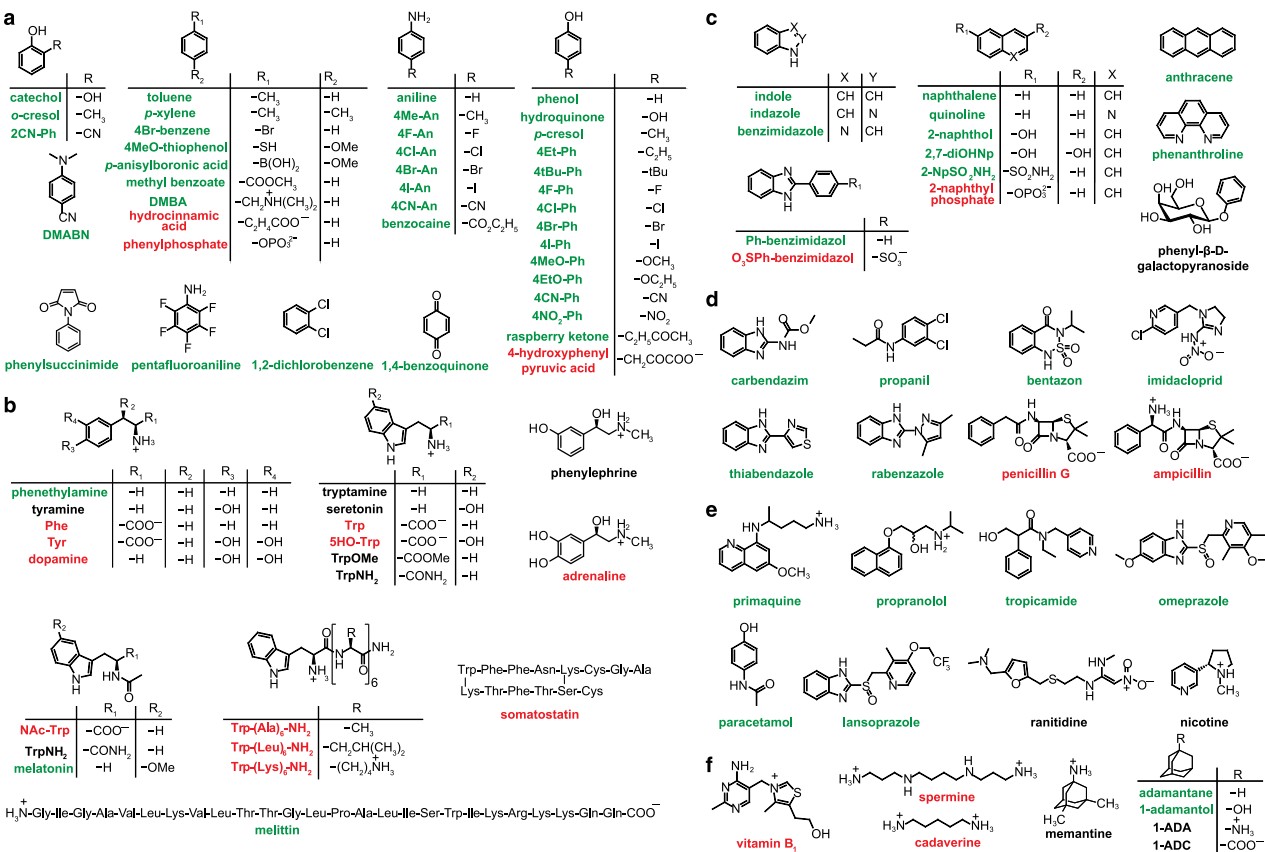

**Fig. 2 Chemical structures of compounds investigated in this study. a** Benzene derivatives; **b** amino acid derivatives and peptides; **c** polycyclic aromatic and heterocyclic compounds; **d** pesticides, insecticides, fungicides, herbicides, and antibiotics; **e** drugs; and **f** vitamin B1 as well as selected non-aromatic compounds, for which the alternative dye-displacement strategy was adopted (see the Supplementary Information). The compounds are represented in their predominant charge state at pH 7. Color code; green: rapidly permeable, black: slowly permeable, and red: impermeable.

measured with **FAR-1** and **FAR-2**, as well as with two different dye-displacement strategies. Reassuringly, the normalized kinetic profiles were superimposable, within error. As an example of very fast translocation kinetics, a representative stopped-flow data set is shown in Fig. 3e for the addition of phenol to **FAR-2** liposomes. It was found that the observed kinetic traces could be well fitted by a monoexponential decay model, affording pseudo-unimolecular rate constants ($k_{obs}$). As expected, the $k_{obs}$ values were found to be linearly proportional to the analyte concentration, at least in a low concentration range (see the Discussion). The observed kinetic rate constants for 28 structurally related, non-charged, and rapidly permeating phenol- and aniline-type analytes were obtained analogously (Supplementary Table 2). The recovered rate constants span more than three orders of magnitude in range. Such data can be utilized to compare the permeation characteristics of different analytes, to uncover structure–property relations, to derive permeation rate constants ($k_p$) and to extract apparent permeability coefficients ($P_{app}$), see the "Discussion."

When compared with PAMPA and Caco-2, our FARMA method offers access to the entire time course of the permeation process, which can be employed for detailed mechanistic investigations of the permeation process. Kinetic investigation of the more complex case of charged analytes illustrate how the FARMA method can potentially be used to obtain novel mechanistic information. Unlike the findings for small, neutral analytes (Fig. 3), the permeation of charged tryptamine, TrpNH2, and serotonin led to significant deviations from monoexponential kinetic traces (see Fig. 3c, d and Supplementary Figs. 6, and 8g). This is in line with translocation models

for charged species, which predict deviations from simple monoexponential kinetics on account of co-transport of counterions or the neutralization of the charge by a prior deprotonation step[29]. The lipid composition of the bilayer membrane can also strongly influence the permeation rates[30,31]. Herein, we tested POPC : POPS compositions ranging from 8 : 1 to 15 : 1 (Fig. 3) and the influence of cholesterol as a membrane component (see Supplementary Fig. 7).

**Spatially and temporally resolved FARMA with GUVs.** For specialized permeation assays, it would be desirable to monitor the permeation kinetics of single membrane-compartmentalized entities instead of the ensemble average obtained with small FAR-encapsulated liposomes. We therefore tested whether a spatially and temporally resolved permeation monitoring is possible with FARMA. To this end, giant unilamellar vesicles (GUVs), which can be studied by conventional fluorescence microscopy, were loaded with **FAR-1** using electroformation (see the "Methods"). To allow direct imaging, we skipped a potential separation step to remove the non-encapsulated **FAR-1** from the buffer medium. Instead, the non-permeating analyte tryptophan was added to the media, which saturates the binding sites of the non-encapsulated **FAR-1** and, thus, quenches the extravesicular emission. The inner compartment of the **FAR-1**-encapsulated GUVs is not accessible to the analyte Trp and, thus, it remains emissive (Fig. 3g) and available for binding of a subsequently added, membrane-permeable analyte. Indeed, when TrpOMe as a rather quickly permeating charged analyte (Table 1) was added to the medium with **FAR-1**-encapsulated GUV, a loss of the fluorescence

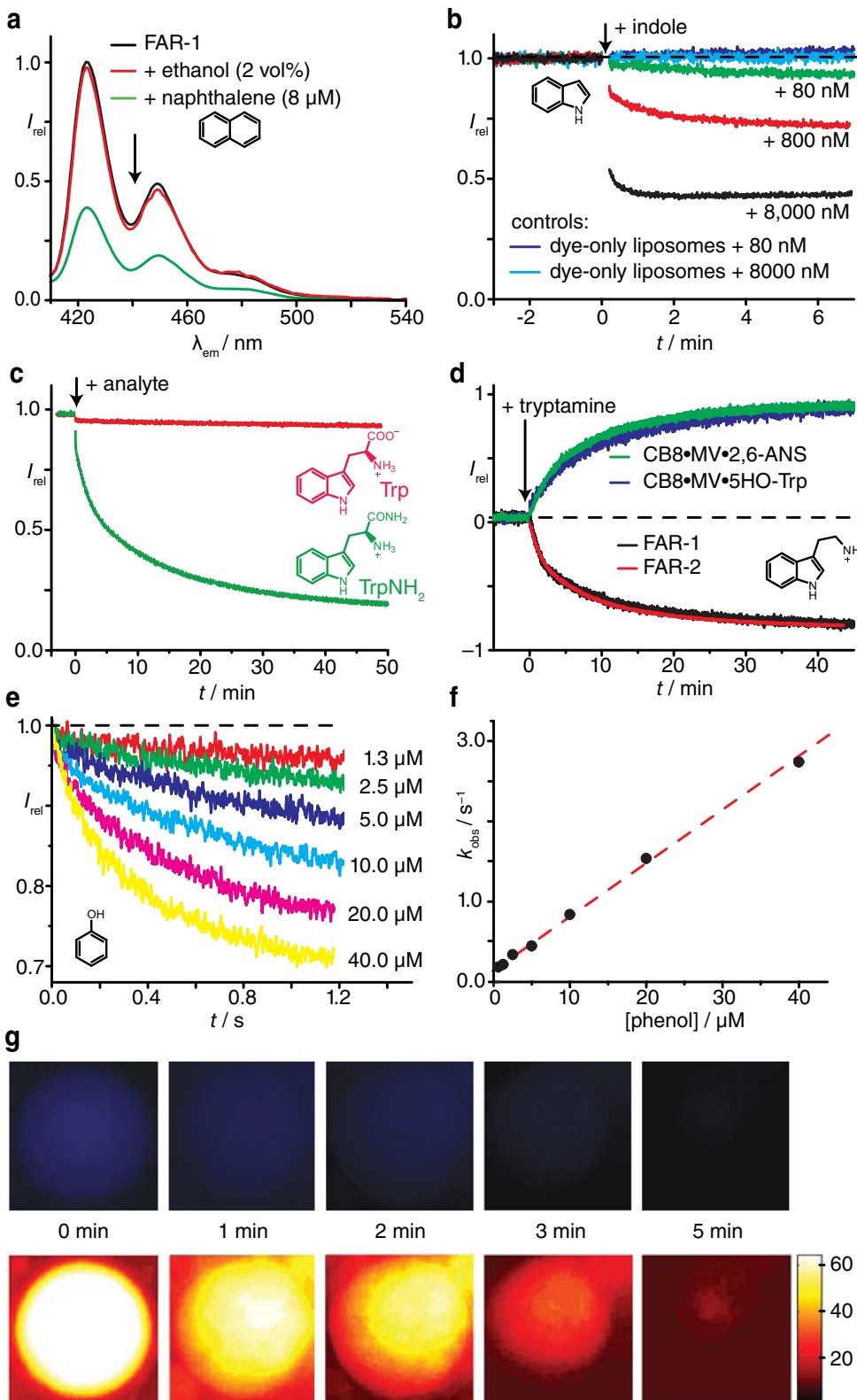

emission from the interior of the GUVs was observed over time, indicating that TrpOMe reaches the GUV-encapsulated **FAR-1** target (Fig. 3g). This sequence of experiments also serves as an independent, visual verification that the FARs are permanently encapsulated inside the liposomes, and that lysis of the membrane upon analyte addition does not occur (see also Supplementary Fig. 8).

**Selective detection of analytes and enzymatic conversions.** Beyond its use for permeation monitoring of drugs and other biorelevant analytes, FARMA opens up new sensing opportunities, overcoming standing issues related to the low selectivity of FARs[26]. For instance, selective detection of permeable species (e.g., TrpOMe) is possible even in the presence of aromatic amino acids such as tryptophan that would, in homogeneous solution,

**Fig. 3 Temporally and spatially resolved FARMA experiments. a** Emission spectra of **FAR-1**-loaded liposomes prior and after addition of naphthalene (ethanolic stock); addition of neat ethanol is shown as the control. **b** Emission intensity of **FAR-2**-loaded liposomes upon addition of indole (aq. stock); control exp. with **D1**-loaded liposomes are shown in blue. **c** Time-resolved translocation monitoring of tryptophan (Trp) and tryptophanamide (TrpNH$_2$), both 8 µM from aq. stock, with **FAR-1**-loaded liposomes (POPC : POPS 8 : 1, 10 mM HEPES buffer, 22 °C). **d** Translocation monitoring of tryptamine (aq. stock, 16 µM) with **FAR-1**- and **FAR-2**-loaded liposomes, and with two membrane-encapsulated indicator displacement ensembles (blue and green, see Supplementary Information). Experiments were conducted with POPC : POPS 8 : 1 liposomes in 10 mM HEPES buffer at 22 °C. **e, f** Kinetic traces from stopped-flow experiments for rapid mixing (1 : 1 v/v) of phenol (aq. stock) with **FAR-2**-loaded liposomes (POPC : POPS 8 : 1, 10 mM HEPES buffer, 22 °C). **f** Plot of $k_{obs}$, from monoexponential fits of kinetic traces vs. phenol concentration. **g** Time series of fluorescence microscopy images of a **FAR-1**-loaded GUV after addition of 5 µL tryptophan methyl ester (TrpOMe, 800 µM stock) to the medium, as real-color images (top) and as intensity-coded images (bottom). Experiments were conducted with POPC : POPS (15 : 1) liposomes at 22 °C.

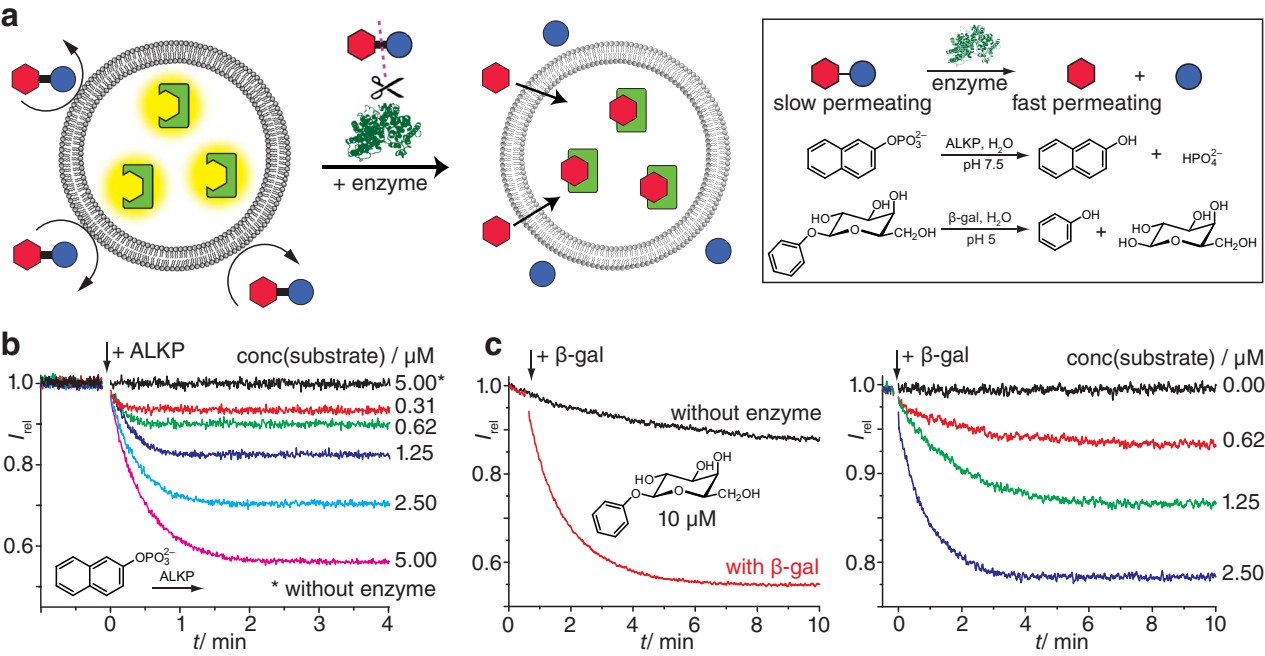

**Fig. 4 Enzyme-coupled FARMA method. a** Schematic operational principle and representative enzymatic reactions. **b** Time-resolved emission of liposome-encapsulated **FAR-1** with the membrane-impermeable substrate 2-naphthyl phosphate and the enzyme alkaline phosphatase (ALKP, 16 µg ml$^{-1}$). Experiments were conducted with POPC : POPS (8 : 1) liposomes at 22 °C. **c** Time-resolved emission of liposome-encapsulated **FAR-1** and the slowly permeating substrate phenyl-β-D-galactopyranoside, with and without the enzyme β-galactosidase (β-gal, 43 µg ml$^{-1}$). Experiments were conducted with POPC : POPS (8 : 1) liposomes at 22 °C.

quench the emission of the FAR (see Fig. 3g). Furthermore, analytes can be also distinguished from each other when their permeation rates are sufficiently different (see, for instance, the examples in Supplementary Figs. 9 and 10).

Several charged, non-permeable analytes can be selectively detected when a suitable enzyme is used in combination with FARMA (Fig. 4a). For instance, negatively charged, impermeable aryl-phosphates such 2-naphthyl phosphate (Fig. 4b) are selectively converted to permeable phenols (e.g., 2-naphthol) upon addition of the enzyme alkaline phosphatase (ALKP). This new compartmentalized variant of a supramolecular tandem enzyme assay[32] allows for the detection of down to 300 nM arylphosphates, while the magnitude of the response is considerably lower in the absence of the protective membrane (Supplementary Fig. 11). Similarly, the affinities of phenol and phenyl-β-D-galactopyranoside for **FAR-1** are comparable, $K_d$ = 1.8 mM and 3.3 mM, respectively, such that they cannot be readily distinguished in a homogeneous sensing format. However, with FARMA, phenol gives an "immediate" signal ($t < 5$ s, see Fig. 3e) whereas phenyl-β-D-galactopyranoside provides a slow response ($t > 5$ min, see Fig. 4c). Upon addition of the enzyme β-galactosidase (β-gal), a fast hydrolysis of phenol-substituted β-D-galactopyranoside occurs[28], upon which the phenol product

quickly permeates through the membrane and binds to **FAR-1** (see Fig. 4a). Besides, information about the enzymatic reaction rate can be derived at the same time from the enzyme-coupled FARMA experiments. The membrane-mediated spatial separation of FARs from the enzymes ensures native functionality and catalytic activity of the enzyme. Finally, with FARMA, enzymatic reaction monitoring is feasible at lower substrate concentration than in homogeneous FAR-based sensing formats.

## Discussion

The FARMA method affords information-rich kinetic permeation profiles as the primary output. These can be directly compared with each other to rank the membrane permeation characteristics of different analytes. The relative permeation characteristics extracted in this way suffice for the majority of envisioned practical applications of FARMA, e.g., when testing drug candidates.

From the permeation screening experiments, analytes can be divided into non-permeating, slowly permeating, and rapidly permeating (see color codes in Fig. 2 and in Supplementary Table 3). Almost all neutral and positively charged species were found to be readily membrane permeable, unless they are polar and large (such as peptides), very hydrophilic (such as

**Table 1 Permeation rates ($k_p$) and apparent permeability coefficients ($P_{app}$) for the permeation of charged analytes through liposomal POPC : POPS (8 : 1) bilayer membranes ($r$ ca. 100 nm),[a] ordered from slowest to fastest permeability coefficient.**

| Analyte | $k_p/(10^{-2}\,s^{-1})$ | $P_{app}/(10^{-6}\,cm\,s^{-1})$ | | |
| --- | --- | --- | --- | --- |
| | This work[b] | This work[b] | Literature[c] | Assay/lipid type |
| Trp[d] | <0.01 | <0.0003 | 0.00041 | End-point analysis/EPC liposomes |
| | | | 1.0[e] | Caco-2 |
| Serotonin | 0.12[f] | 0.004[f] | 1.1 | Aliquot analysis/flat lipid bilayer |
| Ranitidine | 0.39[f] | 0.013[f] | 0.88 | PAMPA |
| | | | 0.49[e] | Caco-2 |
| TrpNH$_2$ | 5.2 | 0.17 | 2.7 | PAMPA |
| | | | 4.3 | Caco-2 |
| | | | 12 | Caco-2 |
| Tryptamine | 5.6 | 0.19 | 5.4 | PAMPA |
| (initial rates) | | | 6.7 | Aliquot analysis/flat lipid bilayer |
| | | | 0.33 | Fluorescence quenching |
| NATA | 31 | 1.0 | 1.9 | PAMPA |
| | | | 2.5 | Caco-2 |
| | | | 2.4 | Caco-2 |
| | | | 0.1 | Fluorescence quenching |
| TrpOMe | 190 | 6.4 | - - - | |
| Memantine | 720 | 24 | 43 | Caco-2 |
| Indole | 3200 | 106 | 32 | PAMPA |
| | | | 57 | Caco-2 |
| | | | 250 | Aliquot analysis/flat lipid bilayer |
| Phenol | 3400 | 112 | 47 | PAMPA |
| 4Cl-aniline | 5000[f] | 170[f] | 42 | PAMPA |
| Aniline | 9400[f] | 310[f] | 76 | PAMPA |
| 4CN-phenol | 50,000[f] | 1700[f] | 17 | PAMPA |

[a]Obtained by the FARMA method with a receptor concentration of ca. 500 μM (500 μM CB8 and 550 μM dye) at 22 °C; entries 4Cl-aniline, aniline, and 4CN-phenol correspond to Supplementary Table 2.
[b]Value obtained by Eq. (1), taking the slopes $k_{obs}/c_{analyte}$ from the linear fit of $k_{obs}$ *vs.* $c_{analyte}$ in the linear range, i.e., $c_{analyte} \leq 20\,\mu M$, unless stated otherwise, 20% error (reproducibility).
[c]Literature values taken from the following references: end-point analysis/EPC liposomes[31], aliquot analysis/flat lipid bilayer[30], Caco-2[42], PAMPA[38,40,43], and fluorescence quenching[18]. For memantine, $P_{app}$ was taken from ref. [44] for Caco-2.
[d]No permeation observed up to 100 μM.
[e]No passive permeation but active transport.
[f]Values from single-point measurements according to Eq. (1) at 40 μM analyte concentrations, except for serotonin (16 μM) and ranitidine (100 μM).

dopamine), or dicationic, such as paraquat (viologen), or the dyes **D1**–**D3** themselves. Representative kinetic traces are shown in Fig. 3 and Supplementary Figs. 5, 9, and 10.

The positively charged neurotransmitters tryptamine, tyramine, and serotonin were shown to pass the membrane within several minutes to hours. Expectedly, the more hydrophilic species permeate more slowly, e.g., phenethylamine > tyramine (Supplementary Fig. 9f) and tryptamine > serotonin (Supplementary Fig. 5 vs. 9g). The catecholamine neurotransmitters dopamine and adrenaline (epinephrine) are very slowly permeating through the biomembrane, whereas the parent catechol is "instantaneously" permeating (Supplementary Fig. 9f).

Almost all anionic species were found to be phospholipid membrane-impermeable (Supplementary Table 2 and Supplementary Figs. 9 and 10), which is in agreement with expectation (e.g., for aromatic amino acid derivatives[33] or ampicillin[34]) and rationalized by the Columbic repulsion between the analyte and the negatively charged bilayer biomembrane. The "hydrophobic anion" 2-adamantyl-carboxylate is a noteworthy exception; it is membrane-permeable, albeit at a slower rate than its non-charged (2-adamantanol) and positively charged (2-ammonium-adamantane) analogs (see Supplementary Figs. 16 and 17). In fact, lipidization of drugs through connection to adamantyl moieties is a known approach to increase their membrane permeability and, thus, bioavailability[35].

To rationalize the permeation rate trends of structurally simple, non-charged aromatic species (Supplementary Table 2), the molecular van der Waals volume (vdW volume, $V_W$) of the analyte and the log$P$ values—a measure for the lipophilicity of an

analyte—were employed as descriptors. For example, the permeability of phenol (log$P = 1.64$) is 20 times lower than that of toluene (log$P = 2.52$), which is in line with lipophilicity differences[36]. Furthermore, an inverse relationship between the vdW volume and the permeability is observed for the subset of *para*-alkylated or halogenated phenols, i.e., the $k_{obs}$ are ordered as H > Me > Et > *t*Bu and F ≥ H > Cl > Br > I. However, interesting exceptions were also observed; 4-*tert*-butylphenol is more lipophilic and smaller than propanil but permeates three orders of magnitude more slowly. Indeed, it has been proposed that highly lipophilic molecules (log$P > 3$) can be retained in the lipid membrane and, therefore, exit the membrane slowly, causing an overall decrease in the observed analyte translocation rate[37].

When attempting to correlate the permeation rates with the log$P$ and $V_W$ values for the whole set of 28 small-molecule aromatics, it becomes immediately obvious that a simplified structure–activity relationship using the lipophilicity and size of the permeating species does not exist (see Fig. 5). Such counterintuitive behavior exemplifies the complexity of the passive diffusion through a membrane for which FARMA can provide useful experimental benchmark data.

Elementary physical parameters, such as permeability coefficients, can be extracted from the full kinetic FARMA data. This kinetic profile → parameter mapping requires some assumptions to be made in regard to the permeation mechanism. We applied a reported permeation model for liposomes[31] to arrive at permeation rate constants ($k_p$) and apparent permeability coefficients ($P_{app}$). The derivation of the mathematical relation (1), linking the fundamental permeability rate constant and

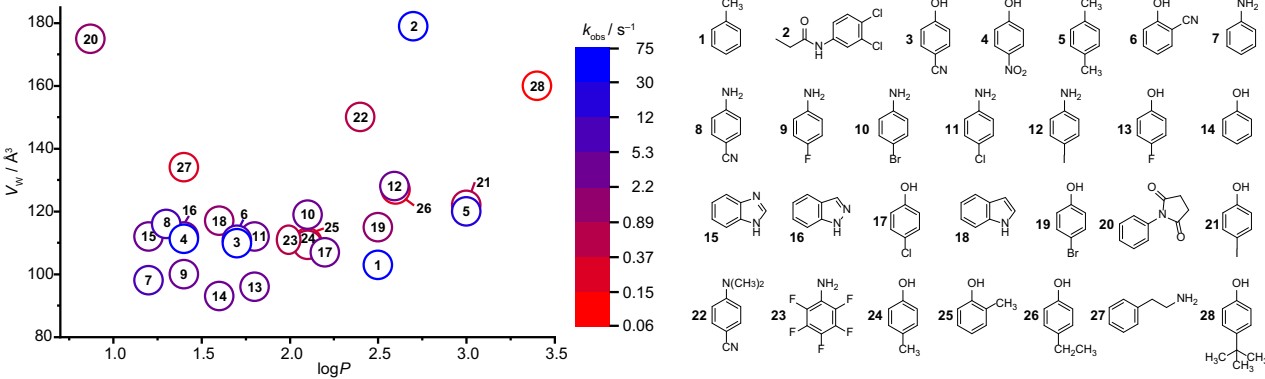

**Fig. 5 Structure-permeability relationships.** Observed permeation rate constants ($k_{obs}$) for small aromatics, e.g., phenols and anilines (color-coded from blue = fastest to red = slowest permeating on a logarithmic scale) correlated to their $\log P$ values (x axis) and their van der Waals volumes ($V_W$, y axis). The numbering of the compounds follows their order of permeation speed from 1 = fastest to 28 = slowest. See Supplementary Table 2 for the numerical $k_{obs}$, $\log P$, and $V_W$ values.

permeability coefficient to experimental measurable $k_{obs}$ rate constants, the known total analyte and total FAR concentration, and the experimentally determined liposome radius ($r$), is shown in the Supplementary Information.

$$P_{app} = k_p \cdot (r/3) = k_{obs}/c_{analyte} \cdot c_{FAR} \cdot (r/3) \qquad (1)$$

The volume-to-surface correction factor $r/3$ accounts for the fact that the observed rates depend on the size, i.e., radius $r$, of the liposomal assembly[31]. Experimentally, the ratio $k_{obs}/c_{analyte}$ can be obtained as the slope of the plot of $k_{obs}$ against $c_{analyte}$, as is shown in Fig. 3f for phenol as the analyte, and in Supplementary Fig. 12 for indole, tryptamine, TrpOMe, and $N$-acetyl tryptophanamide (NATA). The initial-rate method can also be applied, see the Supplementary Information. Moreover, $k_{obs}/c_{analyte}$ can be estimated by single-point measurements at selected $c_{analyte}$ concentration.

Representative permeation rate constants ($k_p$) and permeability coefficients ($P$) obtained through Eq. (1) are listed in Table 1 for selected analytes for which literature $P$-values were available. The $k_{obs}$, $k_p$, and $P_{app}$ values for a series of 28 phenols and anilines is given in Supplementary Table 2.

The obtained permeability coefficients shown in Table 1 compare favorably with ranges of literature values obtained by other methods for neutral analytes with comparable molecular weight[38]. For instance, the $P_{app}$ value (all in $10^{-6}$ cm s$^{-1}$) for indole determined by FARMA in liposomal POPC:POPS bilayer membranes (106) lies within the range of values extracted from PAMPA in synthetic membranes (32)[39], by Caco-2 assay (57)[40], and for a macroscopic bilayer of brain phospholipids (250)[30]. Likewise, when comparing the $P_{app}$ values for other non-charged analytes, a good qualitative agreement is found, e.g., for NATA, where the FARMA value (1.0) lies between that determined by fluorescence quenching (0.1), PAMPA (1.9), and Caco-2 (2.5). For the drug memantine, the FARMA value (24) is close to the reported Caco-2 based value (43). However, for charged and rather hydrophilic substances, our setup yields consistently lower permeabilities than PAMPA and Caco-2. For instance, for tryptamine, our value determined by FARMA in liposomal POPC:POPS bilayer membranes (0.19) comes close to that obtained by fluorescence quenching experiments (0.33) but falls one order of magnitude short to that of PAMPA and other flat lipid membranes (Table 1). Similar findings were made for TrpNH$_2$, ranitidine, and serotonin, which all show an order of magnitude faster permeation under PAMPA and Caco-2 conditions than in our POPC:POPS bilayer membranes. Conversely, for non-charged lipophilic aromatics, we mostly observed faster permeation in our

POPC:POPS bilayer membranes than reported under PAMPA conditions, e.g., compare the series of phenols and anilines. These differences point to a specific mechanistic involvement of the different membrane lipids in the permeation process.

It transpires that the FARMA method affords absolute permeability coefficients, which are comparable with those determined by established methods. Differences may be traced back to the use of flat hexadecane membranes (PAMPA) vs. spherical phospholipid bilayers (FARMA), potential complications arising from transporters (Caco-2), and saturation effects occurring at higher analyte concentration (usually not considered for PAMPA and Caco-2 but uncovered by FARMA). The use of different buffers and pH as well the different lipid compositions and types also contributes to the variations.

In conclusion, the FARMA method allows real-time optical monitoring of the permeation of a large variety of drugs, toxins, and other organic compounds, circumventing the need for labeled analytes[16,17], bypassing methodologies limited to analyte-induced pH jumps[41], circumventing single-point mass-spectrometric detection[15], and complementing alternative assays in membrane research such as PAMPA[10–12] or Caco-2[13]. Important to note, FAR-based membrane assays allow accessing the entire kinetic traces even for the most rapid permeation events. Furthermore, the FARMA procedure can be modified to allow real spatial resolution to microscopically follow analyte uptake. Owing to the use of fluorescence for detection, flexible implementation into microplate and confocal imaging formats are readily performed. Furthermore, different FARs can be adopted that vary with respect to analyte scope, sensitivity, selectivity, and excitation, as well as emission wavelengths. Preliminary experiments have shown that the method is transferable to different lipids such as dipalmitoylphosphatidylcholine, dioleoylphosphatidylcholine, and dioleoylphosphatidylserine, which allows access to the future investigation of permeation rates in dependence on temperature and lipid-phase type.

We therefore contend that FARMA will become a complementary tool both in fundamental and applied membrane permeation research.

## Methods

**Materials.** Analytes, buffers, and lipids were purchased from Alfa Aesar and Sigma Aldrich, and were used as received. Peptides were purchased from BIO-SYNTAN GmbH. Hosts CB7 and CB8 were purchased from Strem. Dyes **D1**–**D3** were prepared according to literature procedures (see ref. [26] and references therein). Dyes 2,6-ANS and DapoxyS, which were utilized for the dye-displacement assay described in the Supplementary Information, were purchased from Invitrogen.

**Preparation of FARs**. The chemosensors **FAR-1**, **FAR-2**, and **FAR-3** were self-assembled from the host CB8 and the dyes **D1**, **D2**, and **D3**, respectively, by dissolving the solid materials together in HEPES buffer (10 mM, adjusted to pH 7.0) to reach 500 μM in CB8 and 550 μM in the dye component (a slight excess of dye was used to ensure full complexation of the host). The dissolution process was assisted by heating to 40–50 °C and the use of a sonication bath.

**Determination of binding constants for FAR-analyte pairs**. Fluorescence titrations were carried out in aqueous HEPES buffer (10 mM, pH 7, 22 °C) unless stated otherwise. To a solution of the FAR (typically at 10–30 μM) was stepwise added a solution of the analyte (typically up to 2 mM) and the fluorescence spectra were recorded. The normalized emissions at 450 nm (**FAR-1**) or at 370 nm (**FAR-2**) were fitted with an equation for a 1 : 1 binding by a least-square fit. The resulting affinity constants are reported in Supplementary Table 1. Representative titration plots are shown in Supplementary Fig. 1. Note that the high CB8-dye binding affinities correspond to a near quantitative degree of host-dye complexation, i.e., quantitative formation of the FAR, such that the subsequent binding of the analyte can be treated independently.

**Liposome preparation and FAR encapsulation procedure**. A solution of 2.5 mg/mL of POPC and 0.33 mg/mL of POPS in chloroform was purged with nitrogen and dried overnight under high-pressure vacuum. The lipid film was rehydrated with 1 mL HEPES buffer (10 mM) containing FAR (0.5 mM, prepared from 500 μM in CB8 and 550 μM in the dye component in HEPES buffer, see above) followed by 13–15 freeze–thaw cycles (freeze in liquid nitrogen, thaw at 40 °C in a water bath). The resulting FAR-loaded liposomes were separated from non-encapsulated species by size-exclusion chromatography (NAP-25 column), while maintaining the same buffer. The absence of non-encapsulated FAR was confirmed by adding a non-membrane-permeable species such as tryptophan to a fluorescence cuvette containing 25 μL of liposomes diluted in 1 mL HEPES (10 mM) buffer. Complete removal of non-encapsulated FAR is indicated by the absence of a significant fluorescence change upon Trp addition. The size of the liposomes ($r \sim$ 100 nm) was measured by dynamic light scattering (Zetasizer Nano from Malvern Instruments).

**FARMA procedure**. In a typical experiment, 20 μL of liposome solution loaded with FAR was diluted in 1 mL HEPES buffer (10 mM, pH 7.0) in a 1 mL quartz cuvette. An emission spectrum was recorded after 10 min of "equilibration time" on a Varian Eclipse spectrofluorometer thermostated at 22 °C with a water bath using $\lambda_{exc} = 400$ nm for **FAR-1**, $\lambda_{exc} = 310$ nm for **FAR-2**, and $\lambda_{exc} = 330$ nm for **FAR-3**. In the time-resolved experiments, the emission intensity at $\lambda_{obs} = 450$ nm for **FAR-1**, $\lambda_{obs} = 370$ nm for **FAR-2**, and $\lambda_{obs} = 370$ and 500 nm for **FAR-3** (the latter is the emerging excimer band for certain **FAR-3•**analyte complexes) was recorded with an averaging time of 0.5 s. Once the signal had stabilized, 8 μL of a 1 mM analyte solution in the same buffer (HEPES, 10 mM) was added ($c_{final} = 8$ μM) to the cuvette and the emission monitoring was continued until no significant change occurred. For single-point experiments, a full spectrum was recorded after a fixed time or several time intervals after analyte addition. Control experiments confirmed that the autofluorescence of each analyte at the given excitation and emission wavelength for a FAR was unnoticeable or small and could be corrected for. It was also confirmed by UV/Vis spectroscopy that the absorbance of the analyte at the given excitation wavelength of the FAR is low (Abs < 0.05), such that inner filter effects are also negligible. Only FAR-analyte combinations were used for FARMA, for which no significant autofluorescence and competitive light absorption of the analyte occurred.

Microplate measurements were performed with a JASCO FP-8500 spectrofluorometer coupled with a JASCO FMP-825 microplate reader accessory in 96-well microplates at ambient temperature, using flat-bottom black microplates with a nonbinding surface. After filling with liposome solution (200 μL), the microplate was placed into the reader and equilibrated for 10 min. Then, fluorescence intensity of each well was recorded, followed by analyte addition and subsequent fluorescence recording at specific time intervals.

**Time-resolved FARMA for rapidly permeating analytes**. Stopped-flow experiments were performed with a Bio-Logic stopped-flow SFM-20 module coupled to a JASCO FP-8500 spectrofluorometer at 22 °C. In a standard experimental setup, 400 μL of a **FAR-2**-loaded liposome solution was diluted in 10 mL of 10 mM buffer in Syringe 1, while Syringe 2 contained phenol at different concentrations, between 2 and 80 μM. Fluorescence measurements were initiated by mixing the contents of the two syringes in equal volumes (total volume = 200 μL, flow speed of 4.5 mL/s) in the stopped-flow chamber, such that the final phenol concentration range for measurement was 1–40 μM. All experiments were carried out in 10 mM HEPES buffer, pH 7.0, at 20 °C. Fluorescence intensities were recorded with an excitation wavelength of 310 nm and emission at 350 nm. For each experiment, measurements from 6 injections were accumulated and the average of these traces was used for data analysis.

**Spatially and temporally resolved FARMA with GUVs**. GUVs were prepared using Vesicle Prep Pro from Nanion Technologies. Specifically, a mixture of a 30 μL POPC solution (25 mg/mL in CHCl₃) and 10 μL POPS solution (5 mg/mL in

$CHCl_3$) was spread as a thin film on ITO-coated glass slides. After the solvent had evaporated and the film had dried, it was rehydrated with 300 mM of sucrose solution containing **FAR-1** and covered with another ITO slide. After 2 h of preparation in Vesicle Prep Pro, GUVs had formed and the suspension was collected. All subsequent measurements were carried out on the same day: a drop of the GUV suspension was pipetted on a glass slide and the formation of GUVs was confirmed by bright-field microscopy (Supplementary Fig. 8). To this suspension, 5 μL of tryptophan (1 mM stock) was added to quench the fluorescence of the **FAR-1** chemosensing ensemble. Fluorescence images of such treated GUVs were taken with a fluorescence microscope (Axiovert 200, Carl Zeiss, filter set 02, i.e., G 365 nm, FT 395 nm, and BP 420 nm), equipped with a digital camera (Evolution QEi monochrome). The first image was taken immediately after the addition of 5 μL of TrpOMe (800 μM stock) to the suspension. Subsequent images were taken at regular intervals (1 min) thereafter. Constant illumination was avoided to reduce potential photobleaching, i.e., the sample was illuminated only when the images were taken. Exposure time and camera settings were constant across all images. Images were analyzed by using the Image J software. To ensure that the apparent decrease in fluorescence over time was not due to photobleaching of the dye, experiments were carried out exactly as described above, except for the addition of TrpOMe. Indeed, in the absence of TrpOMe, no noticeable change in fluorescence intensity was observed (Supplementary Fig. 8).

**FARMA-coupled enzymatic experiments**. The hydrolysis of 2-naphthyl phosphate by alkaline phosphatase (ALKP) from bovine intestinal mucosa (activity of ~2000 units/mg according to supplier) was monitored by emission spectroscopy ($\lambda_{exc} = 420$ nm, $\lambda_{em} = 450$ nm) with **FAR-1** at 37 °C following literature procedures[28]. In analogy to the FARMA procedure, 20 μL of liposome solution loaded with **FAR-1** was diluted in 1 mL HEPES buffer (10 mM, pH 7.5) in a 1 mL quartz cuvette. Once the emission signal had stabilized, the required volume of a 2-naphthyl phosphate solution in the same buffer was added to reach a final substrate concentration between 310 nM and 5 μM in the cuvette. After a ca. 1 min equilibration time, ALKP stock solution was added to reach an enzyme concentration of 16 μg ml⁻¹ in the cuvette. The emission recording was continued until no significant change occurred anymore (after ca. 5 min). The control experiments in homogeneous solution, i.e., in the absence of a protective membrane, were conducted analogously, employing 5 μM **FAR-1** and 5 μM to 30 μM 2-naphthyl phosphate (see also Supplementary Fig. 11a).

The hydrolysis of phenyl-β-ᴅ-galactopyranoside by β-gal from *Aspergillus oryzae* (activity of 8 units/mg according to supplier) was conducted analogously, but at 22 °C[28]. Phenyl-β-ᴅ-galactopyranoside (1 mM) solution was added as substrate to reach a final substrate concentration between 0 and 2.5 μM in the cuvette, and, after ca. 1 min equilibration time, β-gal stock solution was added to reach an enzyme concentration of 43 μg ml⁻¹ in the cuvette. The recording was continued until no significant change occurred anymore (after ca. 10 min). The control experiments in homogeneous solution employed 5 μM **FAR-1** and 10 μM phenyl-β-ᴅ-galactopyranoside (see also Supplementary Fig. 11b).

**Statistics and reproducibility**. Permeation coefficients were based on fitting of real-time fluorescent decay traces with a large number (>100) of data points. No randomization or blinding was used. Reported errors were estimated from the reproducibility, which was checked for representative kinetic experiments by the same and by different operators.

**Reporting summary**. Further information on research design is available in the Nature Research Life Sciences Reporting Summary linked to this article.

## Data availability
Source data for Figs. 3, 4, and 5 are available as Supplementary Data 1, 2, and 3, respectively. All other data are available from the corresponding authors upon reasonable request and are digitally stored on the severs of the home institution.

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

## Acknowledgements

This work was supported by the German Academic Exchange Service (DAAD, F.B.) and the Deutsche Forschungsgemeinschaft (DFG Grants BI-1805/2-1 for F.B., NA-686/11 for W.M.N., and HE 5967/4-1 for A.H.). We thank Mathias Winterhalter for instrument usage, Denisa Hathazia for performing the enzyme-coupled FARMA assay, and Solène Collin for testing reproducibility.

## Author contributions

F.B. and W.M.N. designed the experiments and prepared the manuscript. F.B. prepared the FARs and liposomes, and carried out the FARMA experiments. G.G. performed the complementary dye-displacement-based assays and the GUV studies. A.H. contributed to the fluorescence data analysis by developing the initial-rate method.

## Competing interests

The authors declare no competing interests.
