## [Peer Review File · Communications Biology]

Editorial Note: *Parts of this Peer Review File have been redacted as indicated to maintain the confidentiality of unpublished data*

Reviewers' comments:

Reviewer #1 (Remarks to the Author):

This manuscript describes the use of a cucurbit-[8]-uril dye conjugate system to monitor membrane permeability. While the recognition and sensing capabilities of CD[8] are very well-known, this is an extremely imaginative use of this system. Detecting membrane permeability of small aromatics is very difficult, and the CB[8]-dye system is perfectly suited to this task. The selective and strong binding of guests in a ternary fashion is well-suited for target binding in complex aqueous systems, and the host itself is highly water-soluble and not membrane permeable, so this is a very effective method of monitoring membrane permeability.

While the concept is excellent, the rest of the paper might be even better - the authors have taken the single greatest advantage of this system (continuous in situ readouts) and applied it to a spatiotemporal sensing. This is tricky for any system, and especially for a host:guest indicator assay, and I'm very impressed by how well it works. The enzyme monitoring application is more well-known, but it's a very effective and ingenious method of kinetic analysis.

Overall, this is definitely suitable for CommsBio - it's an elegant sensing system for an important target, the sensitivity is excellent, and the spatiotemporal sensing adds a layer of complexity not usually seen for this type of system. It's well-written, clear, interesting to chemists, biologists, a wide scope of readers. I recommend publication.

Amazingly, I can't find anything to correct, which is very rare for me. This is excellent, and can be published without change.

Reviewer #2 (Remarks to the Author):

In this manuscript, the authors report an assay for evaluating membrane permeability of compounds. The assay utilizes supramolecular fluorescent receptors encapsulated in liposomes. Compounds that penetrate into the liposomes bind to the receptors and quench the fluorescence, thereby reporting its membrane permeability.

The benefits of the assay over other assays are multiple: 1. label-free detection; 2. real-time monitoring of membrane permeation; 3. high sensitivity. The assay is applicable to a wide variety of compounds and allows researchers to obtain not only permeability values, but also kinetic information, that are not easily obtained using other assays. Therefore, the reviewer considers that the work is suitable for publication in Communications Biology.

However, the reviewer requests the authors to address the following two major points to further ensure the generality of the assay and validity of the permeability values obtained via the reported assay before the manuscript is accepted for publication.

1. To demonstrate that the assay can be conducted with liposomes consisting of various lipid compositions, the reviewer request the authors to show that FARs do not leak from liposomes consisting of lipids other than POPC and POPS. As other lipids, dioleoyl-type lipids, such as DOPC, DOPS, or DOPE, are considered to be good choice since they give different degrees of packing of lipids. If the authors further utilize the liposomes for testing membrane permeability of some of the

compounds appeared in the manuscript using FARMAs composed of such different types of lipids, the results would be informative for understanding relationships between permeability of compounds and lipid structures.

2. The difference of Papp values obtained from FARMA and those from PAMPA or Caco-2 assay are, while interesting, a little concerning. These can simply be from difference of assay conditions. PAMPA is generally conducted in PBS (pH 7.2) at 25 °C while the FARMA assay in this paper is conducted in HEPES buffer (pH 7) at unspecified temperature. Especially for cationic molecules, buffer pH and salt conditions might affect to the results. Therefore, the authors should conduct a side-by-side comparison of FARMA and PAMPA (or Caco-2 assay) by themselves in the same conditions to make a conclusion about whether these assay truly give different results for the same compounds, or not.

In addition to the major points described above, there are the following minor points that need to be addressed.

3. Recently, a convenient method for evaluating cell membrane permeability using HaloTag proteins was reported by Kritzer group. (L. Peraro, et al. *J. Am. Chem. Soc.* 2017, 139, 7792-7802. L. Peraro, et al. *J. Am. Chem. Soc.* 2018, 140, 11360-11369.) In the introduction, in addition to the reference 4-7, the authors are recommended to cite these reports as a previously reported permeability assay.

4. On page S-9 of Supplementary Information, there is an incomplete sentence starting from "From these".

5. On page S-12, a structure of tryptamine is mislabeled as "tyramine". This should be corrected to "tryptamine".

Reviewers' comments:

Reviewer #1 (Remarks to the Author):

This manuscript describes the use of a cucurbit-[8]-uril dye conjugate system to monitor membrane permeability. While the recognition and sensing capabilities of CD[8] are very well-known, this is an extremely imaginative use of this system. Detecting membrane permeability of small aromatics is very difficult, and the CB[8]-dye system is perfectly suited to this task. The selective and strong binding of guests in a ternary fashion is well-suited for target binding in complex aqueous systems, and the host itself is highly water-soluble and not membrane permeable, so this is a very effective method of monitoring membrane permeability.

While the concept is excellent, the rest of the paper might be even better - the authors have taken the single greatest advantage of this system (continuous in situ readouts) and applied it to a spatiotemporal sensing. This is tricky for any system, and especially for a host:guest indicator assay, and I'm very impressed by how well it works. The enzyme monitoring application is more well-known, but it's a very effective and ingenious method of kinetic analysis.

Overall, this is definitely suitable for CommsBio - it's an elegant sensing system for an important target, the sensitivity is excellent, and the spatiotemporal sensing adds a layer of complexity not usually seen for this type of system. It's well-written, clear, interesting to chemists, biologists, a wide scope of readers. I recommend publication.

Amazingly, I can't find anything to correct, which is very rare for me. This is excellent, and can be published without change.

We are very grateful to the very positive assessment of Reviewer 1.

Reviewer #2 (Remarks to the Author):

In this manuscript, the authors report an assay for evaluating membrane permeability of compounds. The assay utilizes supramolecular fluorescent receptors encapsulated in liposomes. Compounds that penetrate into the liposomes bind to the receptors and quench the fluorescence, thereby reporting its membrane permeability.

The benefits of the assay over other assays are multiple: 1. label-free detection; 2. real-time monitoring of membrane permeation; 3. high sensitivity. The assay is applicable to a wide variety of compounds and allows researchers to obtain not only permeability values, but also kinetic information, that are not easily obtained using other assays. Therefore, the reviewer considers that the work is suitable for publication in Communications Biology.

However, the reviewer requests the authors to address the following two major points to further ensure the generality of the assay and validity of the permeability values obtained via the reported assay before the manuscript is accepted for publication.

We are happy about the interpretations of Reviewer 2 in regard to the advantages of the FARMA assay method. Two of the below comments in regard to the investigation of lipid structures and the quantitative methodological cross-comparison of permeability methods present projects on their own and are subject to current funded research projects and PhD or postdoctoral studies in the groups, and we can already report pertinent results of the ongoing projects below.

1. To demonstrate that the assay can be conducted with liposomes consisting of various lipid compositions, the reviewer request the authors to show that FARs do not leak from liposomes consisting of lipids other than POPC and POPS. As other lipids, dioleoyl-type lipids, such as DOPC, DOPS, or DOPE, are considered to be

good choice since they give different degrees of packing of lipids. If the authors further utilize the liposomes for testing membrane permeability of some of the compounds appeared in the manuscript using FARMAs composed of such different types of lipids, the results would be informative for understanding relationships between permeability of compounds and lipid structures.

The FARMA method opens indeed a door for the systematic investigation of permeability in dependence on (i) different lipids, (ii) temperature, (iii) phase transitions, and (iv) additives. These studies are under way in our laboratories but we have now emphasized some key findings in regard to the transferability of the method:

- **Investigation of additives:** We show in the manuscript that different lipid composition are in principle possible by adding cholesterol to the POPC and POPS membrane (Fig S7), causing a noticeable alteration of the permeation kinetics of tryptamine as a model analyte.
- **Different lipid compositions:** We have added a statement on page 10 (bottom) pointing to the possibility of preparing reporter-pair loaded liposomes with different lipid POPC:POPS compositions, ranging from 8:1 to 15:1 (our GUV preparations in the manuscript) to pure POPC (ref. 20 in the main text).
- **Use of different lipids:** The FARMA method does also work in DPPC:DOPS and DOPC:DOPS liposomes, the latter corresponding to the composition enquired by the reviewer. A typical set of fluorescence traces is shown for the example of serotonin below (Fig 1); as expected, different permeation rates are obtained, about a factor 2-4 lower for DOPC:DOPS compared to POPC:POPS liposomes. This allows us to perform temperature-dependent measurements, obtain activation energies for the permeation process, and investigate different lipid phases. This extended mechanistic study will be published separately. Nevertheless, we have added in the conclusions on page 18 a clear statement on these experimental possibilities, which define our future work.

2. The difference of Papp values obtained from FARMA and those from PAMPA or Caco-2 assay are, while interesting, a little concerning. These can simply be from difference of assay conditions. PAMPA is generally conducted in PBS (pH 7.2) at 25 °C while the FARMA assay in this paper is conducted in HEPES buffer (pH 7) at unspecified temperature. Especially for cationic molecules, buffer pH and salt conditions might affect to the results. Therefore, the authors should conduct a side-by-side comparison of FARMA and PAMPA (or Caco-2 assay) by themselves in the same conditions to make a conclusion about whether these assay truly give different results for the same compounds, or not.

We have now explicitly stated that the variations in permeation values may also arise from different buffers, pH, lipids and lipid compositions (page 17). As can be seen from the literature survey (Table 1, right column), these values vary largely. Consequently, we have been conservative and concluded that the absolute values obtained by the FARMA method only agree range-wise with literature data. Our idea for medium term is to couple the PAMPA assay with FARs. In detail, we plan to implement FARs into the donor or receiver well of

PAMPA plates, which will allow us to test permeability for the same membrane structures, and, thereby, allow the direct comparison suggested by the reviewer. We have improved the specification of the measurement temperature for the present assays (page 21 in the FARMA procedure, in the Figure captions of Fig 3 and Fig 4, and in the footnote of Table 1.), addressing this comment by the reviewer.

3. Recently, a convenient method for evaluating cell membrane permeability using HaloTag proteins was reported by Kritzer group. (L. Peraro, et al. J. Am. Chem. Soc. 2017, 139, 7792-7802. L. Peraro, et al. J. Am. Chem. Soc. 2018, 140, 11360-11369.) In the introduction, in addition to the reference 4-7, the authors are recommended to cite these reports as a previously reported permeability assay.

We thank the reviewer for pointing out these very relevant publications, which are now cited (ref 8 und 9) in the Introduction.

4. On page S-9 of Supplementary Information, there is an incomplete sentence starting from “From these”. We thank the reviewer for spotting this. It was now removed from the revised SI.

5. On page S-12, a structure of tryptamine is mislabeled as “tyramine”. This should be corrected to “tryptamine”.

We thank the reviewer for spotting this oversight. It is now corrected in the revised SI.

REVIEWERS' COMMENTS:

Reviewer #2 (Remarks to the Author):

The authors addressed all of my concerns raised in the comments. I now recommend this paper to be published in Communications Biology. This new permeability assay is of significant utility in a broad field of chemical and biological science.